# Artificial Intelligence in the Intensive Care Unit: Present and Future in the COVID-19 Era

**DOI:** 10.3390/jpm13060891

**Published:** 2023-05-25

**Authors:** Michalina Marta Kołodziejczak, Katarzyna Sierakowska, Yurii Tkachenko, Piotr Kowalski

**Affiliations:** 1Department of Anesthesiology and Intensive Care, Collegium Medicum Bydgoszcz, Nicolaus Copernicus University Torun, Antoni Jurasz University Hospital No.1, 85-094 Bydgoszcz, Poland; ksierakowska@cm.umk.pl; 2Department of Anesthesiology and Intensive Care, Władysław Biegański Regional Specialized Hospital, 86-300 Grudziadz, Poland; tkaczenko@hotmail.com (Y.T.); pioytrkow@op.pl (P.K.)

**Keywords:** artificial intelligence, intensive care unit, COVID-19, machine learning

## Abstract

The development of artificial intelligence (AI) allows for the construction of technologies capable of implementing functions that represent the human mind, senses, and problem-solving skills, leading to automation, rapid data analysis, and acceleration of tasks. These solutions has been initially implemented in medical fields relying on image analysis; however, technological development and interdisciplinary collaboration allows for the introduction of AI-based enhancements to further medical specialties. During the COVID-19 pandemic, novel technologies established on big data analysis experienced a rapid expansion. Yet, despite the possibilities of advancements with these AI technologies, there are number of shortcomings that need to be resolved to assert the highest and the safest level of performance, especially in the setting of the intensive care unit (ICU). Within the ICU, numerous factors and data affect clinical decision making and work management that could be managed by AI-based technologies. Early detection of a patient’s deterioration, identification of unknown prognostic parameters, or even improvement of work organization are a few of many areas where patients and medical personnel can benefit from solutions developed with AI.

## 1. Introduction—Pre-COVID-19 Era

Artificial intelligence (AI) is considered to be the fundamental technology of the fourth industrial revolution, with global medical agencies signaling its value in constantly evolving medical care [1]. By analyzing huge databases and verifying developed algorithms, it becomes possible not only to diagnose diseases earlier and more accurately but also to implement more personalized care. The Food and Drug Administration (FDA) has approved numerous devices based on artificial intelligence technology, with the list currently including 343 entries [2]. The first product ever approved, a ventilatory effort recorder dating back to 1995 (FDA approval year 1997), was limited in design to monitor a patient’s respiratory rate in addition to generating an audible or visual alarm when an average value did not fall within the operator-defined range [3]. Nowadays, these solutions are essential elements of the ventilator alerting. Over 20 years later, the FDA permitted the marketing of clinical decision-support software analyzing and notifying neurovascular specialists of a potential stroke by computed tomography (CT) imaging [4]. The images still require reassessment by the specialist at a clinical workstation; however, the in-advance notification of the operator could reduce the time to intervention as compared to the usual standard of care.

Numerous institutions have recognized the value of AI not only as an additional tool for accelerated diagnostics, especially in the era of staff shortages, but also in providing patients with more comprehensive care. At the El Camino Hospital in Silicon Valley, predictive algorithms identified patients at a high risk of falling through the real-time comprehensive analysis of hospitalized patients’ data; the generated information about patients at risk was simultaneously transmitted wirelessly to medical staff, which resulted in a 39% fall-rate decrease within six months after the technology was introduced [5]. The technological advancement of medicine would also make it possible to relieve health care systems from bureaucracy, allowing time for medical staff to communicate more effectively with patients and improve the delivery of medical services. Such an approach is being implemented in Partners HealthCare (Brigham, Women’s and Massachusetts General in Boston) in each patient care area to reduce the administrative tasks of medical staff as well as to improve the detection, diagnosis, treatment, and management of diseases [6].

As a result of a significant acceleration in technological development and the influx of more advanced applications, in 2019, the FDA Commissioner Scott Gottlieb outlined a novel review framework for medical devices implementing artificial intelligence technologies [7]. In the announcement, he emphasized the potential of artificial intelligence and machine learning in transforming the delivery of healthcare services. To assert the highest patient safety while providing an advanced level of care, currently approved AI medical devices are characterized by being non-modifiable at the user-level algorithm, which can be modified by the manufacturer after an updated algorithm has been verified. However, as technological advancements are in sight, the FDA anticipates developing a framework for the evaluation of real-world learning and adapting algorithms.

## 2. The Application of AI in the COVID-19 Pandemic

The spread of the SARS-CoV-2 virus overcame barriers of countries and continents by utilizing the possibilities of globalization, forcing international organizations to search for new crisis-management solutions. Little was known about the new disease, and it was necessary to rapidly reduce viral transmission, identify risk factors, optimize management, and reduce the burden on hospitals that faced an influx of respiratory-compromised patients. The initial unknowns were quickly followed by emerging information chaos owing to a plethora of unverified data and their lack of systematization. It was mandatory to fill this niche promptly. International organizations turned for aid to novel technologies. The European Council intensively monitored AI-based technologies that were deployed globally [8]. Most of them were tools for tracing contacts that aimed to contain the infection rate, especially at the early stages of the pandemic. Likewise, AI was implemented in a number of fields, mainly focused on identifying and reducing the spread of the infection (facilitating diagnosis, identifying people with fever, and detection of epidemic outbreaks), supporting research (accelerating gene sequencing, drug design, and prediction of efficacy), and optimization of hospital management (patients and staff allocation) [9,10,11,12]. The objective of achieving collective immunity to mitigate the threat of COVID-19 infections has led to the creation of various AI models focused on enhancing the vaccination process. These models increase the efficacy of vaccine distribution and identify concerns of patients regarding getting vaccinated. In the early phase of vaccination, lack of general knowledge regarding the efficacy and safety of the vaccine spanned a number of concerns that were publicly shared through online social media. Utilizing big data management and AI models, those trends can be identified and addressed by officials who can create targeted messages that address them when provided with accurate information [13].

The intensive work of multidisciplinary groups allowed for the development of the first AI-based technologies that could be implemented in hospitals facing the surge of COVID-19 patients. Support of the decision-making process has paramount importance when managing a large volume of patients, especially the interpretation of numerous radiological examinations to rule out non-COVID-19 causes of respiratory failure and to attempt to allocate patients who may develop a severe course of the disease. From the level of the hospital emergency department, AI-based tools for rapid COVID-19 diagnostics were created, among others, by Mount Sinai Health System (based on chest computed tomography (CT) scans and patient data) and the University of Minnesota, along with Epic Systems and M Health Fairview (chest X-rays) [14,15]. Another type of AI systems, namely software predictors, allow for the identification of hospitalized patients at high risk of clinical events, for example, intubation (COViage) and respiratory failure or low blood pressure (CLEWICU System) [16,17].

The World Health Organization (WHO) signaled that AI could be an important technology for overcoming the current global health crisis and a tool aiding sustainable recovery of populations, with an endorsed series of publications in the *British Medical Journal* aiming at familiarizing medical professionals with the AI technologies [18]. However, some authors pointed out that AI-based algorithms do not have as optimistic results as were hoped for and expected. A systematic review analyzed more than 100 proposed COVID-19 models for diagnosis and forecasting, which proved to be at high risk of statistical error, poorly reported, and overly optimistic [19]. Following this, the authors recommended that none of the models should be used in medical practice until those problems could be resolved. Similarly, another systematic review focusing on AI-based technologies developed for ICU use (11 predictive AI-based diagnostic models, 2 different lung segmentation software (based on deep learning) for prognosis, and 1 optimization of work in the ICU) found the studies to be at a high risk of error due to poor reporting of missing data, poor model validation, a small number of analyzed patients, or lack of inclusion of censored participants [20]. It is mandatory to assert that the demand for rapid-response technology interventions does not hamper the responsible design and use of AI [21]. Utilization of AI-specific reporting guidelines (SPIRIT-AI and CONSORT-AI) could additionally facilitate the development and quality of AI-based applications [22]. Formally, the FDA seeks to oversee and modify the dynamic process of ensuring the safety of patients who can benefit from the implementation of new technologies in their treatment. Technology is advancing rapidly, which also requires changes to systems for reviewing and providing opinions on rapidly evolving medical devices [23].

## 3. Artificial Intelligence in the COVID-19 Intensive Care Unit (ICU)

Patients requiring ICU hospitalization have compromised chances of survival on their own and have to be supported with invasive treatment techniques related to decompensation of the cardiovascular and respiratory systems’ function. Hospitalization in the ICU also requires a significant individualization of patients’ treatment plans and focuses on a scholastic analysis of their whole clinical profile. Comprehensive intensive management significantly increases the time and resources that must be dedicated to patients’ care. Due to the course of the disease and the rapid emergence of large numbers of COVID-19 patients who required respiratory therapy, prone positioning, renal replacement therapy (RRT), or extracorporeal membrane oxygenation (ECMO), ICUs and their staff became swiftly overwhelmed. New ICU stations were created in rapidly adapted departments to provide the possibility of intensive care for as many patients as possible. After the initial resources crisis, when not enough personal protective equipment (PPE), ventilators, and ICU stations were available—a problem that could not be quickly resolved—there was a shortage of competent intensive healthcare professionals able to operate equipment and manage critical patients. In a short time, efforts were made to train non-intensive care personnel, an initiative that usually takes years under normal training conditions. A possible solution arose: to utilize clinically applicable AI-based technologies to support the decisions of healthcare professionals. The development of such technologies requires close interdisciplinary collaboration between AI developers and medical experts, especially because of the amount and implications of the information used by ICU healthcare professionals. Moreover, the role of AI in such a dynamic setting must be innovative in order to not disrupt the process of patient care and clinical decision making (Figure 1).

Researchers started with development of tools that would help to predict the outcomes of the COVID-19 patients and adequately modify management to increase survival and allocate critical resources in the most optimal manner. A number of machine learning models were built to identify predictors of outcomes in the COVID-19 ICU [24]. Based on retrospective (596 patients) and prospective (443 patients) clinical data, AI technology distinguished predictors associated with ICU survival (age, inflammatory and thrombotic activity, and severity of ARDS on admission to the ICU), ECMO therapy (age, pulmonary dysfunction, and transfer from an outside facility), and RRT (interaction of patient age with D-dimer levels on admission and creatinine levels with SOFA score without Glasgow Coma Score). Similarly, a COVID-19 critical care consortium, i.e., ECMOCARD, was established to characterize the evolution of clinical parameters based on data from 633 mechanically ventilated COVID-19 patients in 20 ICUs in Great Britain during the first wave of the pandemic (1 March–31 August 2020) [24]. Using techniques of machine learning and explainable artificial intelligence, complete clinical data within 48 h of ICU admission to death or discharge were analyzed. In critically ill COVID-19 patients, pronation increased the oxygen level in only 45% of patients; those with blood clots or inflammation in the lungs, lower oxygen levels, lower blood pressure, and higher blood lactic acid levels were less likely to benefit from a prone position. Additionally, the effectiveness of the prone position decreased the later it was applied; thus, authors suggested that patients who are unresponsive may be referred to other interventions such as ECMO. The authors created an online tool based on the initial database that could be utilized by everyone free of charge [25]. It must be emphasized, however, that the data were collected during the first wave of COVID-19 pandemic, and they might not be replicable with the course of disease altered by new mutations and management. This prompts continuous data accumulation and the development of comprehensive and preferably international COVID-19-specific databases. Artificial intelligence was also used in early detection and diagnosis, in which AI algorithms may examine chest X-rays, CT scans, and other medical imaging to identify signs of COVID-19 infection. For example, Kuo et al. assessed the performance of an AI model in detecting COVID-19 on chest X-rays in patients with respiratory symptoms from a cohort of 5894 patients across three different continents [26]. The AI model achieved a sensitivity of 79.1% and a specificity of 60.5%. When compared to radiologists from Asia, the U.S., and Europe, by using an independent dataset of 852 positive and negative COVID-19 cases, the AI program achieved a sensitivity of 82.9% and a specificity of 56.8% compared to the radiologists’ sensitivity of 51.6% and specificity of 99.1%. The authors concluded that the study size and scope provided valuable insights into realistic performance expectations for AI systems predicting COVID-19 on chest X-rays and the challenges of creating truly “generalizable” diagnostic AI models even when using an objective standard such as PCR testing. This phenomenon is applicable in numerous AI models and systems, highlighting the need for validation of those novel technologies in clinical practice. 

Another application of AI in COVID-19 intensive care is to predict the risk of more severe illness on initial presentation. Lazzarini et al. presented a machine learning model that could predict severe cases of COVID-19 such as acute respiratory distress syndrome (ARDS) and emphasized various risk factors that significantly impact disease progression [27]. The study was based on a cohort of 289,351 COVID-19 patients. Researchers demonstrated that the machine learning model “Gradient Boosting Decision Tree” achieved the highest performance, with a 40% increase in efficiency compared to another older models. Furthermore, a comparison of the model’s predictive abilities with the predictions of five clinicians indicated that the model is on par with or outperforms the experts in terms of precision and recall. This study introduced a machine learning model that utilizes patient claims history to predict ARDS. The risk factors employed by the model have been extensively linked to the severity of COVID-19 in the specialized literature. The most contributing diagnoses can be easily retrieved from a patient’s clinical history and can be used for early screening of infected patients. The authors concluded that the proposed model could be a promising tool to deploy in a healthcare setting to facilitate and optimize the care of COVID-19 patients. In another study, the predictive models employed learned from historical data to predict the development of ARDS, a severe outcome in COVID-19 cases [28]. Utilizing data from two hospitals in Wenzhou, Zhejiang, China, the researchers identified the features on initial presentation with COVID-19 that were most predictive of later development of ARDS. These features included a mildly elevated alanine aminotransferase, the presence of myalgias, and an elevated hemoglobin, in this order. The AI-based models, which learned from historical patient data from the two hospitals, achieved 70% to 80% accuracy in predicting severe cases. This innovative approach holds significant potential in enhancing patient care and optimizing resource allocation in intensive care settings, particularly amid the ongoing COVID-19 pandemic.

## 4. Challenges and Limitations of AI

The AI-based technologies require a large volume of good-quality data to develop reliable algorithms, as the proposed solutions will only be as precise as the source information. The availability of healthcare data plays a crucial role in the implementation of realistic AI models in healthcare. The increasing digitization of healthcare systems and advancements in data storage and analysis allow for the generation and collection of numerous patient data such as electronic health records (clinical history, treatments, and outcomes), medical imaging, genomic data, wearable device data, and more, providing valuable resources for training AI models. Robust datasets enable the development of more realistic and representative AI models that can better capture the complexities of real-world clinical scenarios. By incorporating data from different patient populations, geographical regions, and healthcare institutions, AI models can account for variations in disease prevalence, demographics, and treatment protocols that bring challenges in the real-world setting. In this way, AI models can identify previously unrecognized patterns, associations, or correlations within the data that may have significant clinical implications. These discoveries can lead to advancements in diagnosis, treatment, and healthcare delivery, ultimately improving patient outcomes, introducing more personalized medicine, and providing a platform for facilitating research and innovation, including drug discovery, disease modeling, and public health interventions. At the same time, the lack of explainability in AI algorithms may undermine trust and raise transparency concerns. Explanations for AI recommendations are crucial for bringing alignment with medical knowledge, as errors in AI systems can have severe consequences, making liability assignment challenging. Clear guidelines and regulations are needed to address legal and ethical implications and ensure accountability. The continuous learning of AI models poses challenges, requiring ongoing monitoring and validation for accuracy and reliability amid evolving healthcare practices. Unfortunately, despite the technological development, due to unconnected health record systems, barriers to the flow and collection of data are everywhere and occur between medical institutions, administrative regions, and countries. Full utilization of AI technologies’ potential will require integrated data analysis systems. The AI algorithms are based on data and rules obtained from specific disease entities, comorbidities, and ethnic and population data and based on the currently available medical knowledge. In order to create universal tools, it will be necessary to include geographically, socioeconomically, and gender-diversified groups. The undertaking of development works will have to take into account an even greater identification of variables that might influence the prognosis but that would not, however, impact the generalization of the algorithms. Even though those criteria would be met, it is mandatory to acknowledge that AI technologies can facilitate the work of medical personnel, but the decision-making process cannot be solely dependent on them. In the case of the emergence of a novel management or a new disease characterized by a different course, the previously created AI system may return incorrect information based on the analysis of heterogeneous data in the context of previously imputed information. AI models should demonstrate robust performance across diverse patient populations, clinical settings, and data sources, allowing for the generalizability of AI systems among various demographics. Additionally, validation of AI models is mandatory to identify and address potential biases and ethical concerns, ensuring the models do not disproportionately favor or disadvantage specific populations and promoting fairness and equity in healthcare delivery.

Typically, the predictions of the currently developed AI models are more optimistic than their clinical performance. This was the case with the EPIC sepsis-alleviation system, which was implemented in numerous hospitals in the United States and performed worse than initial predictions suggested—after an analysis of nearly 40,000 hospitalizations, it only correctly identified 63% patients at their risk of sepsis compared to 76–83% identified by the designer’s predication curve [29].

Another impact of AI on the medical field is ChatGPT. The generative pre-trained transformer (GPT) is a class of language-learning models (LLM) introduced by OpenAI (San Francisco, California) in 2018. ChatGPT has successfully completed the three United States Medical Licensing Exam (USMLE) tests [30]. Additionally, GPT-3.5 (Codex and InstructGPT) has shown human-equivalent performance on a variety of datasets, including USMLE (60.2%), MedMCQA (57.5%), and PubMedQA (78.2%). Although ChatGPT frequently generates notable results, its ability to handle complex real-world questions and situations, particularly in areas such as medicine that have high cognitive demands, remains to be determined [31]. However, in its present state, GPT is subject to mistakes and oversights. It may struggle with elementary tasks such as basic arithmetic or produce subtle inaccuracies that remain undetected without expert examination. Users have noted that GPT can fabricate references when requested to provide them. This raises concerns among educators about the potential for students to be misinformed when depending on the software. 

Cascella et al. asked ChatGPT to create a medical report for a patient in the intensive care unit with details about ongoing therapies, laboratory samples, blood gas analysis parameters, and respiratory and hemodynamic factors in a random sequence [32]. Upon requesting a structured document, ChatGPT managed to accurately organize the majority of the parameters into their corresponding sections, even when they were presented as abbreviations without any context. ChatGPT displayed a remarkable capacity to learn from its errors, assigning the correct section to previously misplaced parameters simply by inquiring whether the parameter was in the appropriate section and without further prompting. A notable limitation involved establishing causal relationships between conditions such as acute respiratory distress syndrome and septic shock. It is important to acknowledge that the information sources may not be up-to-date or extensive enough to accurately determine causality. Furthermore, ChatGPT was not specifically designed for medical inquiries, so it lacks the necessary medical expertise and context to fully grasp the intricate relationships between various conditions and treatments. Nonetheless, ChatGPT exhibited the ability to offer relevant treatment recommendations based on the provided data, although sometimes, the suggestions were general. 

Within the healthcare sector, GPT may take on roles in research, education, and clinical care [33]. In research contexts, it can assist scientists in formulating questions, devising study designs, and collating data. In the sphere of medical education, GPT can act as an interactive reference resource. It can mimic patient interactions, enabling students to sharpen their abilities in gathering medical histories. GPT is even capable of creating initial drafts of progress notes, patient care plans, and other documents that students need to complete for coursework or while on the wards.

## 5. Data Security and Ethical Rights

The AI-based technologies have a number of pitfalls, including reliability, responsibility, privacy, transparency, and security. After the G7 Health Ministers’ Meeting in Oxford on 4 June 2021, announcements were issued on regulations of AI systems’ management [34]. Due to rapid technological development, the need for strengthening advancement and supervision over AI systems in the health sector emerged. The regulations also emphasized the role of international and interdisciplinary collaboration to determine the clinical evaluation process of healthcare AI-based technologies in order to facilitate development between countries. 

During the pandemic, managing law and ethics faced an unprecedented global challenge. Informatization allowed for a previously unparalleled scope of access to personal data, which can be implemented to assemble large and detailed databases for AI-enabled programs. This phenomenon initiated a discussion on data privacy and personal health data ownership since some organizations exchanged patients’ data with technology businesses and startups in order to assert the development of novel and potentially life-saving AI solutions. Concurrently, those issues can jeopardize dataset representativeness and result in the omission of certain populations. While data exclusion due to the selection bias originating from the lack of consent and data anonymization of patient demographics protect individual privacy, they can result in potential bias and inaccuracies of AI-based prognosis, which can in turn affect specific populations. At the same time, it is necessary to create legal norms and regulations that will govern AI technologies, their developers, and users, as one of the problems with implementing AI-based technologies is liability for errors and damages related to its use.

## 6. Conclusions

Developing techniques, creating interdisciplinary teams, training medical staff in the scope of available solutions, adapting IT networks, as well as creating legal regulations should now go hand in hand in order to optimally ensure the development of AI medical personnel support.

It is necessary to create interdisciplinary teams that will include clinicians, people who know AI modeling, IT specialists, lawyers, and people responsible for the ethical development of the above technology. We can gain a great deal as we slowly pursue the possibilities, but the appropriate units must uphold the rule of law, legal responsibility for the use of AI, and more than ever, ensure data safety and the full respect of data confidentiality and patient rights.

Artificial intelligence has yet to reach its full potential; now is the time to consider the limitations and gaps and learn the lessons needed to create the sustainable, resilient, and citizen-centered future of AI.

## Figures and Tables

**Figure 1 jpm-13-00891-f001:**
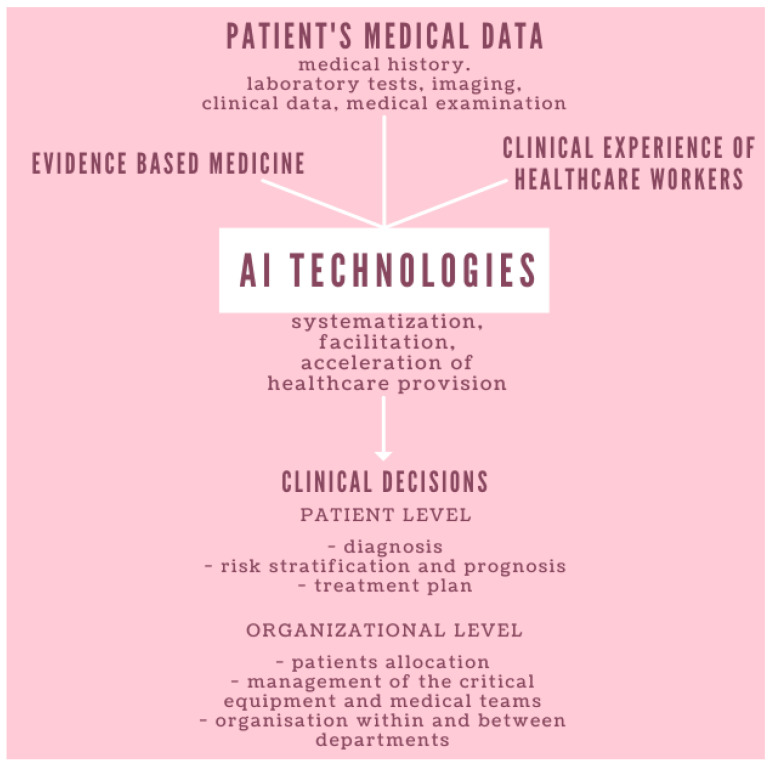
The role of AI-based technologies in healthcare provision.

## Data Availability

Not applicable.

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
