# Peer review of "Artificial Intelligence in the Intensive Care Unit: Present and Future in the COVID-19 Era"

_jpm, 2023, doi:10.3390/jpm13060891_

Round 1

Reviewer 1 Report

The following are suggested to authors

1. Abstract may be improved, relevant to the theme of the paper

2. Related work - containing similar other works may be included

3. The need for validation of AI models/systems for clinical practice is to be discussed 

4. The availability of healthcare data for the implementation of more realistic model may be highlighted

5. also, uncertainities/trustworthiness of AI needs to be mentioned

Author Response

The following are suggested to authors

  1. Abstract may be improved, relevant to the theme of the paper.

REPLY: Thank you for the comment. We improved the abstract, providing more insight on the ICU-oriented care.

  1. Related work - containing similar other works may be included

REPLY: Thank you, we elaborated on nuances raised in further points, related to the specific parts of AI-based technologies. We withheld from citing similar reviews in the field to prevent from over-citations.

  1. The need for validation of AI models/systems for clinical practice is to be discussed 

REPLY: We added lines on validation of AI models within the manuscript: “This phenomenon is applicable in numerous AI-models and systems, highlighting the need for validation of those novel technologies in clinical practice.” and “AI models should demonstrate robust performance across diverse patient populations, clinical settings, and data sources, allowing for the generalizability of AI systems among various demographics. Additionally, validation of AI models is mandatory to identify and address potential biases and ethical concerns, ensuring the models do not disproportionately favor or disadvantage specific populations, and promoting fairness and equity in healthcare delivery.”

  1. The availability of healthcare data for the implementation of more realistic model may be highlighted

REPLY: We added following lines to the manuscript: “The availability of healthcare data plays a crucial role in the implementation of realistic AI models in healthcare. The increasing digitization of healthcare systems and advancements in data storage and analysis allow for generation and collection of numerous patient data: electronic health records (clinical history, treatments, and outcomes), medical imaging, genomic data, wearable device data, and more, providing valuable resources for training AI models. Robust datasets enable the development of more realistic and representative AI models that can better capture the complexities of real-world clinical scenarios. By incorporating data from different patient populations, geographical regions, and healthcare institutions, AI models can account for variations in disease prevalence, demographics, and treatment protocols that bring challenges in the real-world setting. This way AI models can identify previously unrecognized patterns, associations, or correlations within the data that may have significant clinical implications. These discoveries can lead to advancements in diagnosis, treatment, and healthcare delivery, ultimately improving patient outcomes, introducing more personalized medicine, and providing a platform for facilitating research and innovation, including drug discovery, disease modeling, and public health interventions.”

  1. also, uncertainities/trustworthiness of AI needs to be mentioned

REPLY: We added the following explanation: “At the same time, the lack of explainability in AI algorithms may undermine trust and raise transparency concerns. Explanations for AI recommendations are crucial to align with medical knowledge, as errors in AI systems can have severe consequences, making liability assignment challenging. Clear guidelines and regulations are needed to address legal and ethical implications and ensure accountability. The continuous learning of AI models poses challenges, requiring ongoing monitoring and validation for accuracy and reliability amid evolving healthcare practices.”

Reviewer 2 Report

The topic of the presented manuscript is important, relevant, and original. This is a well written manuscript but some issues could be improved.

Introduction seems enough but could achieve a deeper knowledge about state of art.

The chronology is missing.

Discussion section seems confused, needs to be rewritten in some paragraphs.

References. Please, cite references in a Vancouver style (with no “)

Author Response

The topic of the presented manuscript is important, relevant, and original. This is a well written manuscript but some issues could be improved.

Introduction seems enough but could achieve a deeper knowledge about state of art.

REPLY: In the introduction we focus on developments in the pre-COVID-19 era. We elaborate more on the details of AI technologies in dedicated parts of the review, as per suggestion of Reviewer 1.

The chronology is missing.

REPLY: We added a clarification within an introduction of “pre-COVID-19 era”

Discussion section seems confused, needs to be rewritten in some paragraphs.

REPLY: We added some paragraphs and introduced additional corrections to improve clarity.

References. Please, cite references in a Vancouver style (with no “)

REPLY: We changed the references to the Vancouver style.